# Source and Composition in Amino Acid of Dietary Proteins in the Primary Prevention and Treatment of CKD

**DOI:** 10.3390/nu12123892

**Published:** 2020-12-19

**Authors:** Pierre Letourneau, Stanislas Bataille, Philippe Chauveau, Denis Fouque, Laetitia Koppe

**Affiliations:** 1Departement of Nephrology, Hospices Civils de Lyon, Lyon Sud Hospital, 69495 Pierre Bénite, France; pierre.letourneau@chu-lyon.fr (P.L.); denis.fouque@chu-lyon.fr (D.F.); 2Phocean Nephrology Institute, Clinique Bouchard, ELSAN, 13000 Marseille, France; bataille.stanislas@sfr.fr; 3INSERM, INRA, C2VN, Aix Marseille University, 13000 Marseille, France; 4Association Pour l’Utilisation Du Rein Artificiel A Domicile, 33110 Gradignan, France; ph.chauveau@gmail.com; 5University Lyon, CarMeN Laboratory, INSA-Lyon, INSERM U1060, INRA, Université Claude Bernard Lyon 1, 69100 Villeurbanne, France

**Keywords:** chronic kidney disease, amino-acids, plant-based diet, low-protein diet

## Abstract

Nutrition is a cornerstone in the management of chronic kidney disease (CKD). To limit urea generation and accumulation, a global reduction in protein intake is routinely proposed. However, recent evidence has accumulated on the benefits of plant-based diets and plant-derived proteins without a clear understanding of underlying mechanisms. Particularly the roles of some amino acids (AAs) appear to be either deleterious or beneficial on the progression of CKD and its complications. This review outlines recent data on the role of a low protein intake, the plant nature of proteins, and some specific AAs actions on kidney function and metabolic disorders. We will focus on renal hemodynamics, intestinal microbiota, and the production of uremic toxins. Overall, these mechanistic effects are still poorly understood but deserve special attention to understand why low-protein diets provide clinical benefits and to find potential new therapeutic targets in CKD.

## 1. Introduction

Chronic kidney disease (CKD) affects nearly 800 million people worldwide. The kidney has a major role in nutritional homeostasis and partly regulates the amino acids (AAs) pool via their synthesis and degradation. CKD is characterized by a reduction in nitrogenous waste excretion and the accumulation of many organic solutes called uremic toxins (UTs). Many of these UTs are produced by the degradation of dietary proteins and AAs by gut microbiota and appear to accelerate CKD progression and promote a variety of abnormalities, including endothelial dysfunction and insulin resistance. Thus, the composition and balance of gut microbiota in the role of the production of UTs has elicited a recent and important research [1,2].

Experimentally, Brenner et al. [3] showed that a high protein intake induced marked kidney hypertrophy, an increase in intra-glomerular pressure, and hyperfiltration leading to progressive glomerular sclerosis. All observational studies in CKD patients have shown that a high protein intake was associated with CKD progression [4,5]. Based on these observations, a reduction in protein intake can be expected to preserve renal function and is highly recommended by international guidelines for patients with advanced CKD [6].

Although dietary adaptation is a cornerstone of CKD treatment, the mechanistic roles of a low-protein diet (LPD) in CKD pathogenesis and treatment have been partially explored. The interest in the AAs composition of the various animal and plant proteins has emerged after the second world war at a time when the world faced a food shortage [7]. The first aim was to choose proteins so that they become mutually supplementary, and thus to ensure a diet that contains the qualitative and quantitative requirements of essential amino acids (EAAs). However, in industrialized countries, humans rarely face protein/AAs insufficient intake, and for the first time in human history, nutritional excesses and overweight people outnumbers the underweight people. The consumption of protein is around 1.2 to 1.4 g/kg/day, mostly from animal sources, according to the National Health and Nutrition Examination Survey (NHANES) [8]. If a high-protein diet and the keto diet have been suggested to fight against obesity and diabetes, these diets have not been as effective as expected for glycemic control or weight loss in randomized controlled trials (RCT) [9,10]. This calls for a reconsideration of AAs functions in nutrition, now based upon health-related criteria to limit the negative effect on kidney health over time across populations with or at-risk for CKD.

Therefore, recent studies have focused on the quality and diversity of proteins more than on quantity for the prevention and management of CKD. In addition, the increasing consumption of plant foods and vegan/vegetarian diets has remotivated the scientific community to identify its impact on health in the context of CKD [11,12]. In the present article, we review the recent evidence suggesting that the source of proteins (animal versus plant) and composition of AAs specifically affect CKD progression and complications. Because various new functions of AAs on gut microbiota have been recently uncovered, we will focus on recent protein metabolism data on the host/intestinal microbiota balance.

## 2. Clinical Evidence of Plant-Based Diets in CKD Incidence, Progression, and Complications

Plant-based diets -including vegan and vegetarian diets (typically including dairy products and eggs, fruits, plants, grains, and legumes) have recently been considered beneficial for human health [13]. Numerous observational studies in the general population have reported a higher incidence of all-cause mortality, cardiometabolic diseases, and cancer in people who consume greater quantities of red meat [14,15,16], which has been recently challenged. Indeed, in a recent meta-analysis, there was low- or very-low-certainty evidence suggesting that dietary patterns with less red and processed meat intake may result in reductions in adverse cardiometabolic and cancer outcomes [17].

In the context of CKD, another recent meta-analysis of cohort studies linked the daily consumption of plant-based diets and plant proteins with a lower risk of CKD prevalence, incidence, and estimated glomerular filtration rate (eGFR) decline over time [18,19]. However, the review was based on low power RCTs existing in this field with an observation time greater than six months. Most of these trials were conducted in patients with preexisting diseases that predispose to CKD [15,20] as summarized in Table 1 and Table 2. It is important to mention that diet investigations are associated with several biases, and it is important to consider that the association between socio-economic level and animal-protein intakes might influence the results of observational studies.

Therefore, the new international guidelines of the Kidney Disease Outcomes Quality Initiative (KDOQI) did not recommend plant protein over animal protein in CKD due to lack of evidence and because overall data strength was found to be low. Yet, they do recommend a plant-based pattern in CKD patients for its potential benefits on body weight, blood pressure, and net acid production but not for renal protection [6].

## 3. Presumed Mechanisms of Plant-Diets Beneficial Effects on CKD Progression

Preliminary experimental and human studies reported that independently of quantity, plant proteins have different renal effects from animal protein.

There appears to be a beneficial effect on renal hemodynamics with a reduced glomerular hyperfiltration and a decrease in albumin clearance during an equal protein intake regardless of the presence of associated fiber. The mechanism is not yet elucidated but could involve insulin-like growth factor 1 (IGF1) and/or prostaglandins [37] effects on glomerular filtration rate (GFR). The progression of chronic nephropathy in older rats was markedly delayed by replacing casein with soy protein [38]. The different AA content between plant and meat could be the explanation of the observed benefit and will be discussed below.

There is extensive literature on the benefits of plant-diet in CKD progression and complications [11,12,18]. Based on these studies, we have summarized on Figure 1 the mechanisms of plant-based diets on CKD progression. However, not all pathways are fully understood and the effect of plant based-diet on metabolic parameters such as insulin sensitivity or hypertension must be consolidated in large cohorts. Indeed, only few adequate trials have investigated the benefits of plant-based diets on metabolic complications. For example, acidosis improvement is well demonstrated but the effects of plant diets on potassium, uremic toxins or bone parameters need further assessment.”

Firstly, plant-diet regimen brings several beneficial nutrients. It includes a higher fiber intake, which induces a better weight control and improves serum lipid profile [39,40]. A plant-based diet is also associated with higher consumption of mono or polyunsaturated fatty acids, an improvement in lipid profile [41], and a weight reduction [42]. There is also a better control of metabolic acidosis, equivalent to oral sodium bicarbonate supplements on the preservation of eGFR, and a decrease in systolic blood pressure [34,35,36,43,44,45]. Blood pressure control and volume overload are also improved by these diets because they generally contain a lower sodium intake [46]. Concerning phosphorus, these diets decrease serum phosphorus on the one hand because phosphate of plant origin is present as phytate and is less absorbed in the gut [33,47], and on the other hand because they avoid processed food industry products enriched in inorganic phosphate. A plant-based regimen is also associated with a decrease in inflammation and the risk of mortality in CKD patients [48,49].

Secondly, in plant diets, protein quantities are usually reduced and can be assimilated with an LPD that has been associated with a delay in dialysis start [50,51]. Some of the major UTs are produced by intestinal microbiota from diet AAs: Indoles as indoxyl sulfate (IS) and indole-3-acetic acid (IAA) from tryptophane, p-cresyl sulfate (PCS) from tyrosine or phenylalanine, trimethyl amine oxidase (TMAO) from choline or L-carnitine. One of the hypothetical mechanisms of reducing protein intake and increasing plant-based diet in CKD is the capacity to reduce UTs production. Indeed a relationship has been reported between the quantity of protein intake and plasma levels of UTs (such as IS and PCS) and urea have been observed [52]. In this condition, a vegetarian diet (with less precursor of UTs) showed a two-fold reduction in major gut-derived UTs such as PCS and IS in adults with normal renal function [53] and consistent results in patients on maintenance hemodialysis [54]. A short-term (12-week) randomized crossover trial reported that patients with CKD who followed a vegetarian very low-protein diet exhibited a decrease in these UTs and a modification of intestinal microbiota composition [55,56]. The increased intake of fibers in plant-diets allows a modification of colic microbiota towards bacteria that decrease inflammation, bacterial translocation, and the production of gut-derived UTs [57,58,59,60,61].

## 4. Limitations of Plant-Diets in CKD

LPD and plant-based diets share possible caveats or preconceived ideas that may limit their implementation.

### 4.1. Denutrition/Malnutrition

Nowadays, it is well established that LPD is beneficial for the preservation of kidney function and retarding the need for maintenance dialysis [6]. The main reluctance for its implementation is linked to a possible fear for increased protein-energy wasting (PEW) and sarcopenia. Concerning sarcopenia, Garibotto et al. shown that LPD and very LPD (VLPD) are safe in response to an adapted muscle protein metabolism [62]. Concerning PEW, a report of 16 controlled trials listed by Kalantar-Zadeh et al. did not show PEW [18]. In a recent meta-analysis of randomized controlled trials over more than 2000 patients, Hahn et al. did not find a significant modification of nutritional measures when dietary management was performed by trained dietitians. Indeed, the mean final body weight was 1.4 kg higher in patients receiving a VLPD compared to a LPD. The risk for wasting during a VLPD was 0.6%, a magnitude that is not different from a normal diet (0.4%) [51].

The potential hazardous risk of a vegetarian diet is also discussed. This is partly based on beliefs that, as compared to animal proteins, proteins of plant origin would be of lower quality, less complete, poorly digested, and not sufficient to meet daily protein targets in hemodialysis patients. Numerous studies have confirmed the nutritional safety of vegetarian diets in non-dialysis CKD patients [31,63,64]. In addition, the same holds true for maintenance dialysis patients following LPD with plant protein source and higher caloric intakes compared to patients following a LPD with animal protein diet [31]. A number of studies confirmed that is possible to achieve a target of 1–1.2 g protein/kg/day as recommended in hemodialysis patients with a vegetarian diet [63,64].

### 4.2. Bone Disorder

Adequate calcium, protein, and vitamin D intakes are needed for optimal bone health. A plant-based diet is supposed to contain lower calcium with poorer bioavailability. It appears that the absorption of plant-calcium is as efficient or better than that of milk (i.e., 20–40%) if the plant does not have high oxalic acid nor phytic acid concentration. Some plants such as soy or beans can contain as much as 100 mg/100 g of calcium, but overall plants are less concentrated in calcium than dairy products [65,66]. Therefore, ensuring an adequate calcium intake from plants or via supplementation is mandatory with a vegan diet but not in an ovo-lacto-vegetarian diet.

Plant-based diets tend to contain less aminoacids that could lead to osteoporosis. A systematic review confirmed this hypothesis by showing no adverse effects on bone mineral density and hip fracture occurrence with lower compared to higher total protein intake [67]. One RCT comparing isoflavone-rich soy protein versus various animal-based proteins did not show abnormal bone health [68]. One prospective study found that plant diet improved bone mineral density in the general population [69]. Furthermore, a cross-sectional study found an association between plant food consumption and improvement in bone mineralization markers [70]. A meta-analysis including 18 studies found an inverse association between the consumption of fruit and the risk of postmenopausal osteoporosis [71]. Some authors suggested that these reassuring results can be explained through the reduction in BMI, an increase in beneficial phytochemicals, and microbiota modification due to plant dietary patterns [72]. This point has never been specifically explored in CKD patients and needs further studies. However, in the last meta-analysis, which included 17 RCTs and 1459 participants, it was observed that VLPD reduced serum phosphate and parathyroid hormone (PTH) levels suggesting an improved mineral and bone health [73].

### 4.3. Hyperkalemia

Another important question related to plant-based diets is the high potassium content for some of them [11] and the potential risk of hyperkalemia in CKD patients. A recent international recommendation indicates a lack of robust studies assessing potassium intake requirements in CKD patients [6]. Authors highlighted the difficulty of conducting reliable studies due to several reasons such as treatments used (angiotensin-converting enzyme inhibitors, potassium binders), volume status, acid-base status, possible gastrointestinal symptoms, as well as the difficulty of assessing dietary potassium intakes and their poor correlation with serum potassium [74].

In the general population, subjects consuming more potassium develop less CKD [73], and less hypertension [75]. Furthermore, Aburto and al [76] showed that increasing potassium intake reduces blood pressure in known hypertensive patients. In patients with CKD, there are also several studies in favor of a potassium-rich diet that tends to be similar to a plant and fruit diet. However, most of them are retrospective and based on surrogate criteria of 24-h kaliuresis to assess dietary potassium intake. Therefore, they do not establish a direct causal link between increased potassium intake and a reduction in the risk of progression of cardiovascular or renal diseases. Several studies suggest that increased kaliuresis is associated with a lower occurrence of dialysis initiation, a slower progression of CKD and proteinuria, and death [77,78]. It should be noted that in these two studies, the beneficial effect seems to be less significant in patients with more severe CKD (i.e., <60 mL/min eGFR in Nagata et al. [78] and <45 mL/min in Smyth et al. [77]). There are few interventional studies in advanced CKD patients. Tyson et al. reported no hyperkalemia episode during a two-week DASH regimen in patients with CKD stage 3–4 [79]. Goraya et al. showed no association between plant-based diet and hyperkalemia after a five-year follow-up in patients with CKD stage 3–4 [36]. A study of with 11 patients with an eGFR <30 mL/min/1.73 m2 taking a vegan diet did not disclose an increase in serum potassium at three months [80]. On the other hand, plant-based diets, by its fibers content, improve gut transit, reduce constipation, leading to an increased potassium elimination into the feces [81].

Overall, it seems overrated to accuse plant-based diets for high potassium content since high potassium intake occurs as well with highly processed foods of animal origin [82]. Obviously it is possible to reduce the potassium content of foods of plant origin by appropriate cooking. An ongoing RCT of approximately 500 patients with CKD stage 3–4, who will receive either oral potassium supplementation with potassium chloride or potassium citrate versus placebo will answer the question whether potassium alone (i.e., not associated with a plant diet) can lead to better kidney outcomes [83].

## 5. Does the AA Composition Influence the Progression of CKD and Its Complications?

For decades, various health benefits have been attributed to protein restriction in patients with CKD, such as favorable metabolic effects, reduction in proteinuria and uremic symptoms, and improvement in insulin sensitivity [84]. Beside the CKD area, new emerging studies have suggested that a LPD can increase longevity in many species and can retard age-related diseases such as cancer, type 2 diabetes [85], and dyslipidemia/fatty liver disease [86]. Following these observations, several studies have tried to decipher which particular AAs are required to induce the systemic responses to a LPD. In addition, it is tempting to check if specific dietary AA restrictions (one or several AA) are both sufficient and necessary to improve the metabolic response to a LPD. The challenge to apply these specific diets during CKD appears important and complex because (1) diet must ensure that EAAs are sufficient if dietary protein intake is reduced; (2) the recommendation of EAAs requirement is not consensual (Table 3) and not specific for CKD patients; (3) the transposition of experimental to human studies must be carefully interpreted because of EAAs are species-specific and they differ between humans and rodents; (4) the analysis of actual dietary intakes is difficult in human and can be different than prescribed. Serving sizes and food selection vary, making it difficult to estimate single-nutrient and -AA intakes.

### 5.1. Plasma Concentrations of Amino Acids in the Context of Plants and Animal Diets

Both animal and plant proteins are made up by about 20 common AAs. Nine AAs—histidine, isoleucine, leucine, lysine, methionine, phenylalanine, threonine, tryptophan, and valine—are not synthesized by mammals and are therefore dietarily essential or indispensable nutrients. These are commonly called essential AAs (EAAs). The branched-chain amino acids (BCAAs) leucine, isoleucine, and valine belong to the nine EAAs. The proportion of these AAs varies as a characteristic of a given protein, but all food proteins—except for gelatin—contain some of each. For example, EAAs content of plant-based protein isolates such as lupin (21%), oat (21%), and wheat (22%) are lower than animal-based proteins (casein 34%, milk 39%, whey 43%, and egg 32%) and muscle protein (38%). AAs profiles differ among plant-based proteins with leucine contents ranging from 13.5% in corn protein to 5.1% in hemp, compared to 7.0% in eggs, 9.0% in milk, and 7.6% in muscle protein. Methionine and lysine are typically lower in plant proteins such as rice and beans (1.0 and 3.6%) compared with animal proteins (2.5 and 7.0%) and muscle protein (2.0 and 7.8%, respectively) [87]. Furthermore, proteins of animal origin contain more aromatic amino acids (AAAs) (tyrosine, tryptophan, and phenylalanine). Although plant and animal proteins differ in their AAs profiles, digestibility, and availability [11], these differences do not seem to be clinically relevant for the healthy adult general population with a varied and sufficient diet [88]. Indeed, the plasma AAs profile and in particular EAAs [89,90] of plant-based diets and animal-based diets is almost the same, except for a lower lysine-to-arginine ratio [91]. However, CKD patients have strikingly different plasma concentrations of AAs from those of healthy subjects [91,92,93]. For example, BCAAs concentration is lower in CKD patients and non-EAAs are higher. These variations may be related to modified protein and AAs metabolism, to low energy and protein intakes, or to AAs and protein losses due to peritoneal dialysis or hemodialysis. Further specific studies are needed to determine if plant diets can affect, in the long term, the plasma concentration of AAs in CKD patients.

### 5.2. Influence of Specific Amino Acids on Renal Hemodynamics

Early mechanistic studies suggested that not all proteins and AAs equally induce glomerular hyperfiltration. This finding has been proposed because, in a preliminary study in which 10 healthy volunteers consumed equivalent amounts of animal or plant protein for three weeks, consumption of plant protein resulted in a reduction in renal plasma flow and increased renal vascular resistance [37,94]. No difference was found in plasma AAs levels following the two types of protein loads but glycine, alanine, methionine, and lysine were less abundant in plant proteins, suggesting a potential mechanism for the beneficial effects of plants diet [37,94]. In another seminal study, five adult volunteers were infused on separate occasions with BCAAs or a mixture of non-EAAs and EAAs. BCAAs infusion caused a moderate renal vasoconstriction, a slight increase in GFR, whereas the AAs mixture induced a significantly higher increase in GFR and a state of renal vasodilation [95]. An experimental study had 5/6 nephrectomy rats submitted for five weeks to either a control diet (18% casein protein) or diets containing 8% casein supplemented with 10% of a mix of BCAAs or AAAs. These AAs (BCAA and AAA) displayed seemingly opposite effects. The AAAs stimulated renal plasma flow and to a lesser extent GFR, possibly via an effect on the production of bioactive molecules such as 5’AMP-activated protein kinase (AMPK), which is a sensor of cellular energy homeostasis. The BCAAs supplementation was associated with a higher level of fibrosis in the kidney and an increase in plasma free fatty acids pointing to an abnormality at the level of energy metabolism [96]. BCAAs have been shown to impact metabolism by modulating phosphoinositide 3-kinase (PI3K)-AKT signaling and its downstream transcriptional regulation, in particular of the transcription factor KLF15 (Kruppel-like factor 15). In a CKD mice model, an invalidation of KLF15 was associated with an increased renal fibrosis [97]. The KLF15 downregulation and renal fibrosis are reversed with dietary protein restriction in mice. These results suggest that KLF15 may play a role in suppressing renal fibrosis and could contribute to the benefits reported during dietary protein restriction and BCAAs restriction. By contrast, we showed in a preliminary work that a specific restriction in aromatic AAs intake in CKD mice mitigated inflammation that plays a major role in the progression of renal damage [98]. Therefore, studies investigating underlying mechanisms of AAs in renal progression are lacking and remain to be done.

### 5.3. Influence of Specific Amino Acids on Uremic Toxins Generation and Intestinal Microbiota

There is a growing recognition of the role of diet in modulating the composition and metabolic activity of the human gut microbiota, which in turn can impact health. Probably independently of fiber intake, the composition of AAs from plant origin may influence the production of UTs and potentially alleviate symptoms associated with the accumulation of UTs.

#### 5.3.1. Aromatic Amino Acids (AAAs)

The transformation of tryptophan to indolic uremic toxins (UTs) IS and IAA requires bacterial tryptophanase from *Citrobacter, Escherichia*, and *Proteus*, among others. Indolic UTs (like IS and indole-3-acetic acid) are ligands of the transcription factor aryl hydrocarbon receptor (AhR), also known as the dioxin receptor. AhR activation is known to mediate cardiotoxicity, vascular inflammation, and renal dysfunction. Breakdown of tyrosine and phenylalanine by intestinal anaerobic bacteria, such as *Bacteroides, Bifidobacterium, Lactobacillus, Enterobacter,* and *Clostridium* generate phenols such as p-cresol. P-cresol is absorbed in the gut and metabolized to PCS in enterocytes and liver. In a preliminary work, we have shown that a specific restriction in the intake of AAAs in a CKD mice model lowered UTs concentrations as efficiently as a LPD [98]. However, in a small cohort of hemodialysis patients, Brito et al. failed to show an association between dietary tryptophan intake and IS levels. A weakness of this study was the fact that the tryptophan intake was only evaluated by a 24-hr dietary recall performed on three different days [99]. The amount of indolic UTs produced could not only be related to dietary tryptophan but also to the indole production by microbiota tryptophanases and through the liver transformation of indole into IS and IAA [100].

More recently, in a large cohort of 223 hemodialysis patients, using the characterization of the gut microbiome, serum, and fecal metabolome, permutational multivariate analysis of variance analysis revealed that the food categories (such as plant, red meat, fish, fruit, egg, etc.) did not have a significant impact on this UTs profile and microbiome. If the composition of diet was evaluated by a food frequency questionnaire of 117 items, the quality of protein was not evaluated, as well as the quantity of tyrosine or tryptophan [101]. Overall, the bacterial metabolism of AAAs is both multifaceted, interconvertible, and non-fully explored in CKD.

#### 5.3.2. Sulfur-Containing Amino Acids (SAAs)

Although a number of sulfur compounds are metabolized by the gut bacteria, sulfide is generated into the human large intestine by two principal routes: By the action of sulfate-reducing bacteria on inorganic sulfur (sulfate and sulfite) and through the fermentation of sulfur-containing amino acids (SAAs). The chief sources of dietary sulfur are inorganic sulfate and the SAAs methionine, cysteine, cystine, and taurine [102]. Dietary SAAs intake has been reported to preserve CKD progression in patients and experimental disease models [103,104]. To date, it is thought that diet acts to alter the relative abundance (or diversity) of gut microbes, which can then lead to changes in gut microbial metabolite production. Indeed, sulfur compounds increase the abundance of sulfur-reducing bacteria, such as *Escherichia coli* and *Clostridium* spp., in the colon, which increases the generation of hydrogen sulfide (H_2_S). However, Lobel et al. [105] reported that diet can post-translationally modify the gut microbial proteome (S-sulfhydration), which can alter microbial metabolite production to drive changes in renal function. Interestingly, they demonstrated that SAAs can increase sulfahydrated of tryptophanase in *Escherichia coli* and induce a reduction of indole production in CKD mice. These results are in contradiction with the potential beneficial effect of plants because methionine and cysteine are proportionately slightly lower in legumes. Similarly, urea can contribute to a post-translational modification of proteins via increased O-glycosylation and could potentially modify bacterial activity [106]. Increased O-glycosylation of mucins strongly and negatively affects the mucus layer cohesive properties in the gut lumen [107]. Further, protein glycosylation is central to the physiology of intestinal bacteria like *Bacterioides fragilis* [108]. However, the production of urea is not linked to specific AAs. Finally, methionine could potentially influence methylation by providing methyl groups (CH3). Epigenetic alterations such as DNA methylation could potentially provide explanations for altered gene expressions in CKD and in bacterial metabolism [109]. Therefore, the interaction between diet, microbial metabolism, and posttranslational protein regulation must be explored in CKD.

#### 5.3.3. L-carnitine and Choline

Dietary choline and L-carnitine are precursors of TMAO. Choline is metabolized by gut microbiota to trimethylamine (TMA), which is absorbed and oxidized by hepatic flavin monooxygenase-3 (FMO) into TMAO. A phosphatidylcholine challenge (ingestion of two hard-boiled eggs for example) also increases plasma TMAO in humans [110]. This response is suppressed by the administration of antibiotics, which modify the microbiota or by the use of a synthetic diet with a choline analog such as 3,3-dimethyl-1-butanol in mice [111].

The link between TMAO concentrations and renal function has been demonstrated by Tang et al. who observed in healthy male mice fed with a choline-supplemented diet an increase in collagen deposition and tubulointerstitial fibrosis [112]. By contrast, choline-deficient diets have also been associated with hepatic steatosis and disrupt the intestinal barrier [113].

In addition, L-carnitine, which can be produced via nutritionally derived lysine and methionine has been proposed as a therapeutic agent to improve mitochondrial energy metabolism. Carnitine transports free-fatty acids into the mitochondria to maintain mitochondrial function. A meta-analysis has demonstrated that L-carnitine can improve metabolic parameters and survival but the results are not conclusive in the context of CKD [114]. In vitro, cultured myoblast incubation with L-carnitine restored the IS-induced decrease in mitochondrial membrane potential. In a CKD mice model, L-carnitine did not affect plasma IS concentration, suggesting no effect on intestinal metabolism of tryptophan. However, L-carnitine can limit the induction of myostatin and atrogin-1 expression in CKD mice and improve exercise and mitochondrial status by increasing peroxisome proliferator-activated receptor-gamma coactivator (PGC)-1alpha expression (a transcriptional coactivator that is a central inducer of mitochondrial biogenesis) [115].

Furthermore, it seems that there is a much lower production of TMAO during plant-diet because it is mainly induced by a diet rich in red meat. It is also noted that changes in microbiota caused by a vegetarian/vegan diet induce resistance to the production of TMAO when these precursors (choline, L-carnitine) are ingested [111,116,117,118,119].

Despite an overwhelming amount of evidence demonstrating correlations between choline and L-carnitine, and their metabolite TMAO, in terms of disease onset, there are conflicting data about their relationships [120]. Some inconsistencies occur about L-carnitine showing protective effects against metabolic diseases and therefore cast doubt on TMAO’s involvement. First, the way l-carnitine is administered is important and may explain some of the contradictory results. L-carnitine administered intravenously could bypass gut microbial conversion of L-carnitine to TMAO and provide beneficial mitochondrial effects. Secondly, the knowledge of metabolism of choline and L-carnitine is rather limited. The development of a faster and simpler Ultra Performance Liquid Chromatograph- mass spectrometer (UPLC-MS)/MS method for the simultaneous determination of TMAO, TMA, and other choline metabolites in different types of biological samples may help establishing the effect of dietary factors on TMAO metabolism [121]. Finally, not only TMAO, but also TMA, have been described as deleterious, and the latter has been proposed to be involved in cardiovascular pathology, at least in vitro [122].

### 5.4. Amino Acids Composition and Metabolic Complications

Disorders of glucose homeostasis affect approximately 50% of CKD patients even in absence of diabetes. Over 30 years ago, experimental studies reported that treatment of uremic patients with an LPD improved insulin resistance [123]. Until today, there has been no identified picture of the specific role of AAs on metabolic perturbation in the general population and even less in CKD patients. On one hand, the restriction of EAAs such as the BCAAs [124] and SAAs [125] have been shown to be responsible for the systemic effects of a LPD. On the other hand, others have demonstrated that the altered non-EAA metabolism is necessary and sufficient for explaining the beneficial effects of LPDs [86]. For example, the increase of SAAs and H2S generation on metabolic effect is unclear. H2S is a double-sided sword: At low levels, it exerts beneficial effects, but at higher concentrations it can damage pancreatic beta cells and glucose homeostasis, in animal models. By contrast, decreased levels of plasma H2S levels were reported in diabetic patients [126]. Feeding mice and humans with a diet specifically deprived of BCAAs was sufficient to improve glucose tolerance and body composition to the same extent as a LPD [127]. There are evidences that BCAAs and mostly leucine stimulates the expression of mitochondrial biogenesis genes like PGC1 and plays a leading role in mammalian target of rapamycin complex 1 (mTORC1) activation that is involved in the process of protein synthesis and energy metabolism. However, leucine-stimulated mTOR signaling is partly attenuated in skeletal muscle of chronically uremic rats [128]. In healthy older males, ingestion of 6 g BCAA increases myofibrillar protein synthesis rates during the early postprandial phase [129]. Using ketoacids that are produced as a result of BCAA degradation (removal of the amino group) by the enzyme branched-chain amino-acid aminotransferase can be an attractive strategy to efficiently stimulate muscle protein synthesis in CKD [130]. A systemic deprivation of threonine and tryptophan are both adequate and necessary to improve metabolic parameters as LPD in obesity context [131]. One proposed mechanism is that reduction of threonine and tryptophan and LPD promotes such metabolic remodeling and health by the liver-derived hormone fibroblast growth factor 21 (FGF21). However, the role that FGF21 plays in pathophysiology of CKD remains elusive. We have demonstrated that the metabolism of tyrosine, which is converted to PCS by the host, triggers insulin resistance [132]. However, in the other way, a decrease of AAAs in CKD mice is not able to restore insulin sensitivity [98]. The contradictory results can be explained by the simultaneous increase in other non-protein/AA nutrients such as certain carbohydrates that are required for LPD effects [133].

## 6. Conclusions

In summary, the analysis of the literature presented in this review shows a large benefit in the adoption of LPD and plant-based diets in CKD patients. These diets may have an impact on CKD progression and also in the management of CKD-associated metabolic complications. The increase in fibers intake associated with plant-based diet might result in potentially reduced risk of cancer, metabolic and vascular diseases, and death in this population. However, additional intervention trials are required in particular to assess the risk of hyperkalemia even if there is no clear evidence to support limiting plant-based foods in normokalemic patients. The evolution of the society and the realization of clinical studies on the topic in the coming years will be helpful to validate and motivate CKD patients to adopt this diet in safe conditions.

The evolution of the concept of protein restriction to targeted restriction of AAs is now emerging. (Figure 1) A better understanding of the role of each AA in the progression of CKD will make it possible to give targeted nutritional advice and to open up new therapeutic avenues notably interacting with the intestinal microbiota. However; the human gut microbiota is highly variable and structurally distinct from experimental animals, setting up more challenges to the discovery and validation of novel microbial targets to reduce UTs production. Given that many metabolic pathways are maintained in microorganisms, it is possible to identify several functionally redundant yet phylogenetically distinct species, making it difficult to identify the bacterial origin for one specific UT. By contrast, the existence of certain genes does not necessarily mean that a downstream metabolite will be fully produced. The addition of post-translational modifications in the processes partially explains the difficulty to find ways to reprogram the metabolic output of the gut microbiome with diet. However, all of these challenges should not limit scientific efforts to deepen the relationships between the intestinal microbiota, diet, and renal function. On the contrary, the complexity of the interactions allows a larger possibility of new therapies on the microbiota. In particular, the emergence of new powerful bioinformatics analysis tools and using shotgun metagenomic sequencing allow characterization of non-bacterial components of the microbiota, provide information about function and open new hopes. Additionally, note that most of the data available on intestinal microbiota come from analysis of stool; however, there are several niches inside the gastrointestinal tract and stool is only one. Finally, much is yet to be learned and discovered and the coming years are likely to be highly informative and promising. 

## Figures and Tables

**Figure 1 nutrients-12-03892-f001:**
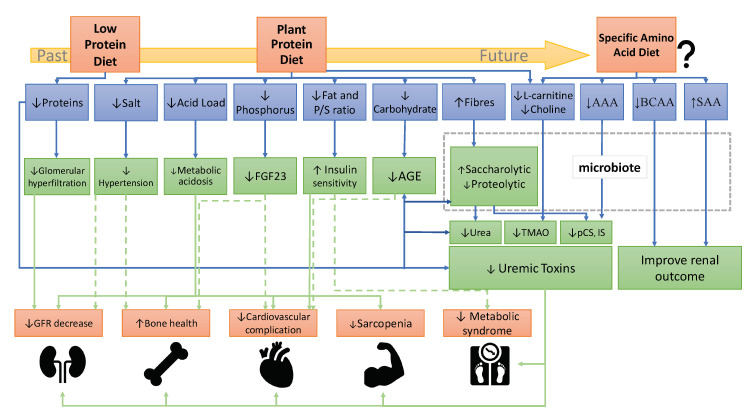
Recommendation evolution of a dietary protein in the prevention and treatment of CKD. Abbreviations: AAA, aromatic amino acids; AGE, advanced glycation end-products; BCAA, branched-chain amino acids; FGF23, fibroblast growth factor 23; GFR, Glomerular filtration rate; IS, indoxyl sulfate; pCS, p-cresyl sulfate; P/S ratio FA, polyunsaturated/saturated ratio of fatty acids; SAA, sulfur-containing amino acids; TMAO, trimethylamine-N-oxide.

**Table 1 nutrients-12-03892-t001:** Observational studies analyzing plant protein or plant-based dietary pattern and chronic kidney disease (CKD) incidence or progression.

Study	Type	Intervention/Assessment	Population	Follow Up	Outcomes	Results/Comments
Lin et al. 2010 [21]	Observational	FFQ: Salt, animal fat, several nutrients	n = 3348 General population	11 years	eGFR decline Microalbuminuria	↗ animal fat consumption was associated with ↗ risk of microalbuminuria (OR, 1.51; 95% CI, 1.01 to 2.26) ↗ β-carotene intake appeared protective against eGFR decline (OR, 0.56; 95% CI, 0.40 to 0.78)
Gutiérrez et al. 2014 [22]	Observational	FFQ: Plant-Based pattern score	n = 3972 CKD G3-G5	6 years	All-cause mortality Kidney failure	↘ risk of mortality with higher adherence of plant-based diet (HR, 0.77; 95% CI; 0.61 to 0.97, *p* < 0.05) No negative association between plant diet and risk of ESKD
Chen et al. 2016 [23]	Observational	Dietary interview: Plant protein–total protein ratio and total plant protein intake	n = 14,866 including 728 patients with CKD	6.2 to 8.6 years	All-cause mortality	↗ plant protein–total protein ratio is associated with a significant ↘ risk of death in CKD population with eGFR < 60 mL/min/1.73 m² (HR, 0.77; 95% CI, 0.61 to 0.96)
Haring et al. 2017 [15]	Observational	FFQ: Plant protein intake	n = 11,952 General population	23 years	Incident CKD	↘ risk of developing CKD with ↗ consumption of nuts (HR, 0.75; 95% CI, 0.65 to 0.85, *p* < 0.001), low-fat dairy products, or legumes (HR, 0.83; 95% CI, 0.72 to 0.95, *p* = 0.03)
Herber-Gast et al. 2017 [24]	Observational	FFQ: consumption of whole grains, fruit, and plants	n = 3787 General population	15 years	eGFR decline Albuminuria creatinine ratio	No association on multivariate model of fruit and plant intakes with changes in renal function or albuminuria creatinine ratio
Asghari et al. 2018 [25]	Observational	FFQ: Lacto-vegetarian, traditional Iranian, and high fat, high sugar dietary pattern	n = 1630 General population	6.1 years	Incident CKD	↘ CKD incidence (OR, 0.57; 95% CI, 0.41 to 0.8, *p* = 0.002) with ↗ adherence to the vegetarian dietary pattern
Liu et al. 2019 [26]	Observational	FFQ: vegan, ovo-lacto vegetarian, or omnivore diets	n = 55,113 General population	Cross-sectional	Prevalence of CKD	↘ CKD in vegan diet adherent (OR, 0.87; 95% CI, 0.75 to 0.97, *p* = 0.018) and ovo-lacto vegetarian diet adherent (OR, 0.84; 95% CI 0.77 to 0.88, *p* < 0.001)
Jhee et al. 2019 [27]	Observational	FFQ: nonfermented and fermented plant and fruit	n = 9229 General population	8.2 years	Incidence of CKD Incident proteinuria	↘ CKD incidence (HR, 0.86; 95% CI, 0.76 to 0.98, *p* < 0.05) and proteinuria incidence (HR, 0.68; 95% CI, 0.59 to 0.78, *p* < 0.05) with highest versus lowest intake of nonfermented plants
Kim et al. 2019 [28]	Observational	FFQ: Healthy pro-vegetarian diet to less healthy	n = 14,686 General population	24 years	Incident CKD eGFR decline	↘ CKD incidence with higher adherence to the healthy plant-based diet (OR, 0.86; 95% CI, 0.78 to 0.96, *p* = 0.001) and pro-vegetarian diets (OR, 0.9; 95% CI, 0.82 to 0.99, *p* = 0.03) ↘ eGFR annual decline with healthy plant-based diet (OR, −1.46; 95% CI, −1.50 to −1.43 *p* <0.001)
Oosterwijk et al. 2019 [16]	Observational	FFQ: protein intake, including types and sources of protein	n = 420 Type 2 diabetes population	Cross-sectional	Prevalence of CKD	↗ intake of vegetable protein is associated with ↘ prevalence of CKD in higher tertile (OR, 0.47: 95% CI, 0.23 to 0.98, *p* = 0.04)
Saglimbene et al. 2019 [29]	Observational	FFQ: fruit and plant intake	n = 8078 Adults on maintenance hemodialysis	2.7 years	Mortality	In the hemodialysis population, ↗ consumption of fruit and plant is associated with ↘ all-cause (OR, 0.8; 95% CI, 0.71 to 0.91, *p* = 0.002) and non-cardiovascular death (OR, 0.84; 95% CI, 0.70 to 1.00, *p* = 0.14)

Studies were selected in PubMed database with a follow up over than 6 months. Abbreviations: CKD, chronic kidney disease; ESKD, end-stage kidney disease; eGFR, estimated glomerular filtration rate; FFQ, Food Frequency Questionnaire; OR, odd ratio; *p* for *p*-value.

**Table 2 nutrients-12-03892-t002:** Interventional studies analyzing the effect of plant protein or plant-based dietary pattern on CKD complications.

Study	Type	Intervention /Assessment	Population	Follow Up	Outcomes	Results/Comments
**INTERVENTIONAL STUDIES**						
Fanti et al. 2006 [30]	Randomized controlled trial	Isoflavone-containing soy-based nutritional supplements (soy group) or isoflavone-free milk protein (control group)	n = 25 ESKD on chronic hemodialysis with systemic inflammation	8 weeks	Impact on inflammatory markers and nutrition markers	↗ serum isoflavone levels associated with ↘ CRP (HR = −0.599, *p* = 0.02) and ↗ albumin (HR = 0.522, *p* < 0.05) No significant decrease of CRP between the two groups but a trend in the soy protein group
Soroka et al. 1998 [31]	Randomized cross-over trial	Plant protein diet versus animal protein diet	n = 9 CKD G3-G4	1 year	eGFR decline	Failed to find a difference between APD versus VPD but it was underpowered and short trial A better degree of compliance with caloric, protein, and phosphate intakes
Tabibi et al. 2009 [32]	Randomized controlled trial	Soy flour (14 g of soy protein) versus usual diet	n = 40 ESKD on peritoneal dialysis	8 weeks	Serum lipid profile	↘ serum Lipoprotein A concentration in the soy protein group (*p* < 0.05)
Moe et al. 2011 [33]	Cross-over trial	Vegetarian versus meat diet comparison	n = 8 CKD G3-G4	7 days	Impact on phosphorus homeostasis	↘ phosphorus serum concentration (*p* = 0.02) and ↘ FGF23 (*p* = 0.008) in the vegetarian diet
Goraya et al. 2013 [34]	Randomized controlled trial	Oral NaHCO3 compared with fruit and plant diet with a controlled arm	n = 106 CKD G4 with metabolic acidosis	1 year	Metabolic acidosis	Fruit and plant diet are as effective as oral bicarbonate to improve metabolic acidosis (19.9 versus 19.3 mM; *p*= 0.01), without an increase of hyperkaliemia risk.
Goraya et al. 2014 [35]	Randomized controlled trial	Oral NaHCO3 compared with fruit and p diet with a controlled arm	n = 108 CKD G3 A > 1	3 years	Urine excretion of angiotensinogen eGFR decline	Fruit and plant diet are as effective as oral bicarbonate decrease angiotensinogen urine excretion (*p* < 0.05) and preserve eGFR (*p* < 0.01) versus usual care
Goraya et al. 2019 [36]	Randomized controlled trial	Oral NaHCO3 compared with fruit and plant diet with a controlled arm	n = 108 CKD G3-4 A > 1 Nondiabetic	5 years	Metabolic acidosis, eGFR decline and CVD risk factors	Fruit and plant diet are as effective as oral bicarbonate to correct metabolic acidosis (*p* < 0.01), eGFR decline (−10.0, 95% CI −10.6 to −9.4 mL/min/1.73 m2 versus −18.8, 95% CI −19.5 to −18.2 mL/min/1.73 m2 in usual care group), *p* < 0.01.) and was better than bicarbonate to reduce systolic blood pressure (*p* < 0.01) It was more effective to lower low-density lipoprotein and increase serum vitamin K1

Only randomized controlled trials were selected in PubMed database. Abbreviations: APD, animal protein diet; CRP: C reactive protein; CVD, Cardiovascular disease; CKD, chronic kidney disease; ESKD, end-stage kidney disease; eGFR, estimated glomerular filtration rate; FGF23, fibroblast growth factor 23; NaHCO3, sodium bicarbonate; VPD, plant protein diet; HR, hazard ratio.

**Table 3 nutrients-12-03892-t003:** Recommendation of estimated amino-acids requirement From WHO (2007) [87].

Amino Acid	Requirements, mg/kg per day	mg/g Protein (1)
Histidine	10	15
Isoleucine	20	30
Leucine	39	59
Lysine	30	45
Methionine	10	16
Cysteine	4	6
Phenylalanine plus tyrosine	25	38
Threonine	15	23
Tryptophan	4	6
Valine	26	39
Total	184	277

(1) Mean nitrogen requirement of 105 mg nitrogen/kg per day (0.66 g protein/kg per day).

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
