# Peer review of "Source and Composition in Amino Acid of Dietary Proteins in the Primary Prevention and Treatment of CKD"

_nutrients, 2020, doi:10.3390/nu12123892_

Round 1

Reviewer 1 Report

The review by Letourneau and co-workers present valuable state-of-the-art insights on the possible role and expected outcomes of protein-based diets in nutrition-based prevention of CKD. The review is interesting for both nephrologists, nutritionists and microbiome-oriented microbiologists.

Major comments:

  • In section 4 of the review, the authors discuss potential limitations of plant-diets in CKD. At this point, the review would be more in balance if potential limitations of low-protein diets for CKD patients would be discussed as well.
  • The conclusion pictures a rather pessimistic outlook towards possibilities for microbiome modulation of UT production. While it is true that functional redundancy in the gut microbial ecosystem can complicate data interpretation and mechanistic unravelling, the authors should try to end with a more promising outlook taking into account the powerful arsenal of bioinformatics tools, systems biology and in vitro simulation approaches that is emerging from the microbiome field.

Other comments:

  • Table 1: Whenever available, p values of the cited effects should be mentioned. Only in this way, a minimum of critical reflection and interpretation of the reported results is possible by the reader.
  • Figure 1 is a bit controversial. The increase or decrease of the metabolic and clinical parameters included here is only scientifically justifiable when accompanied by literature references. Certainly in case of clinical parameters (e.g. hypertension, insulin sensitivity, etc.), it is well known that readouts can vary substantially in the one or the other direction between different study designs, study cohorts, etc.
  • The entire manuscript should be thoroughly checked for typographical errors:

L48: ‘industrialezed’, instead of ‘industrialized’

L181: ‘hyperkaliemia’, instead of ‘hyperkalemia’

L410: ‘microorganisms’, instead of ‘microorganism’

L319: In ‘Clostridium spp.’, the ‘spp.’ should not in italics.

L323: ‘Escherichia Coli’, instead of ‘Escherichia coli

L320 and following: the ‘2’ in the chemical notation H2S should be in subscript

Author Response

Reviewer 1:

The review by Letourneau and co-workers present valuable state-of-the-art insights on the possible role and expected outcomes of protein-based diets in nutrition-based prevention of CKD. The review is interesting for both nephrologists, nutritionists and microbiome-oriented microbiologists.

Thank you for this positive commentary.

  • “In section 4 of the review, the authors discuss potential limitations of plant-diets in CKD. At this point, the review would be more in balance if potential limitations of low-protein diets for CKD patients would be discussed as well.”

We have added a paragraph about concerns onPEW and sarcopenia in LPD diet with 2 new citations L147 to L158 and did minor modifications between L159 to L167

“Nowadays, it is well established that LPD is beneficial for the preservation of kidney function. The main reluctance for its implementation is linked to a possible increase protein-energy wasting (PEW) and sarcopenia. Concerning sarcopenia, Garibotto et al. shown that LPD and very LPD are safe thanks to the adaptation of muscle protein metabolism [63]. Concerning PEW, 16 LPD controlled trials listed in Kalantar-Zadeh et al. review did not report PEW[64]. In a recent meta-analysis of randomized controlled trials to assess the influence of protein restriction on non diabetic CKD patients, Hahn et al did not find a significant modification of nutritional measures with dietary management performed by trained dietitians. Indeed, the mean final body weight was 1.4 kg higher with very LPD compared to LPD. The risk for wasting during very LPD is 0.6 %, a magnitude that is not different from normal diet (0.4 %). [52].

Vegetarian diet is associated with the same concern particularly in dialysis patients in which dialysis-associated protein loss could be a concern. This is partly based on certain beliefs such as those proteins of plant origin would be of lower quality, less complete, poorly digested, and not sufficient to meet daily protein targets for hemodialysis patients. Numerous studies have confirmed the nutritional safety of vegetarian diet in CKD dialysis and non-dialysis patients [32,65,66]. Also, the same is true for maintenance dialysis patients following LPD with vegetable protein diet with higher caloric intakes compared to patients following LDP with animal protein diet [32]. Different studies have observed that is possible to achieve the targets of 1-1.2 g/kg/day protein as recommended in hemodialysis patients with a vegetarian diet [65,66].”

  • “The conclusion pictures a rather pessimistic outlook towards possibilities for microbiome modulation of UT production. While it is true that functional redundancy in the gut microbial ecosystem can complicate data interpretation and mechanistic unraveling, the authors should try to end with a more promising outlook taking into account the powerful arsenal of bioinformatics tools, systems biology and in vitro simulation approaches that are emerging from the microbiome field”

Thank you for these remarks. We have added a positive conclusion to the manuscript. See L 441.

“However, all of these difficulties should not limit scientific efforts to deepen the relationships between the intestinal microbiota, diet and renal function. On the contrary, the complexity of the interactions allows a large wide possibility of new therapies on the microbiota. In particular the emergence of new powerful bioinformatics analysis tools and using shotgun metagenomic sequencing allow characterization of non-bacterial components of the microbiota provide information about function andopen new hopes. Also, note that most of the data available on intestinal microbiota come from analysis of stool; however, there are several niches inside the gastrointestinal tract and stool is only one. Finally, much is yet to be learned and discovered and the coming years are likely to be highly informative and promising.”

  • “Table 1: Whenever available, p values of the cited effects should be mentioned. Only in this way, a minimum of critical reflection and interpretation of the reported results is possible by the reader”

We add odd ratio or hazard ratio with p value when possible.

  • “Figure 1 is a bit controversial. The increase or decrease of the metabolic and clinical parameters included here is only scientifically justifiable when accompanied by literature references. Certainly in case of clinical parameters (e.g. hypertension, insulin sensitivity, etc.), it is well known that readouts can vary substantially in the one or the other direction between different study designs, study cohorts, etc.”

We understand that the figure may seem controversial. However, it is based on several articles detailed in Part 3 and is mainly based on three reviews (11 Carrero et al. Nature Reviews. Nephrology 2020; 12 Chauveau et al. Nephrol. Dial. Transplant. 2018 and 18-Kalantar-Zadeh et al. Nutrients 2020). Moreover, its goal is to draw a global vision on the effects of plant diets and to put into perspective the evolution of nutritional intervention in renal disease and to ask if nutritional intervention on specific amino acid diets can lead to a new era in renal disease.

We have moderated the figure in the manuscript.

“Based on these studies, we have tried to summarize the mechanisms of plant-based diets on CKD progression in Figure 1. However, not all pathways are fully understood and the effect of plant based-diet on some clinical and metabolic parameters like insulin sensitivity, hypertension must be consolidated in large cohorts. Indeed, there are not always excellent trials that have investigated the benefits of plant-based diets on different metabolic complications. For example, the impact of acidosis is well demonstrated but the effect of potassium, uremic toxins or bone parameters need further study to investigate these hypotheses.”

  • “The entire manuscript should be thoroughly checked for typographical errors:

We corrected theses sentences as requested by the reviewer

  • L48: ‘industrialezed’, instead of ‘industrialized’
  • L181: ‘hyperkaliemia’, instead of ‘hyperkalemia’
  • L410: ‘microorganisms’, instead of ‘microorganism’
  • L319: In ‘Clostridium spp.’, the ‘spp.’ should not in italics.
  • L323: ‘Escherichia Coli’, instead of ‘Escherichia coli’
  • L320 and following: the ‘2’ in the chemical notation H2S should be in subscript”

Reviewer 2 Report

Dear authors, the current manuscript is a very interesting review on the recent data on the role of decreasing protein intake, their vegetable nature of their origin, and specific AAs on kidney function and metabolic disorders. The information gathered in this work is really valuable for whoever is interested in the topic, although much further research is needed to understand the underlying mechanisms.

I have only few comments to mention.

My main comment is that, according to the abstract, I would expect some comments/conclusions about plant-based diets and plant-derived proteins at the conclusions of the manuscript. Now, they seem quite unconnected to me.

Line 69: You say "Numerous observational studies...", while you cite only one reference. Please add some more.

Line 72: I think that the sentence "In the latest recent meta-analysis.." would better start with a word showing the antithesis with the previous findings, like "however".

All the best for your work

Author Response

Dear authors, the current manuscript is a very interesting review on the recent data on the role of decreasing protein intake, their vegetable nature of their origin, and specific AAs on kidney function and metabolic disorders. The information gathered in this work is really valuable for whoever is interested in the topic, although much further research is needed to understand the underlying mechanisms.

Thank you for this positive commentary.

  • My main comment is that, according to the abstract, I would expect some comments/conclusions about plant-based diets and plant-derived proteins at the conclusions of the manuscript. Now, they seem quite unconnected to me.”

We have modified the conclusion as requested by the reviewer.

“In summary, the analysis of the literature presented in this review find large benefit in the adoption of LPD and plant-based diets in CKD patients. These diets may have an impact on CKD progression and also in managing the metabolic complications of CKD. The increase of fiber intake associated with plant-based diet might result in potentially reduced risk of cancer, metabolic and vascular diseases, and death in this population. However, many gaps exist in knowledge and additional intervention trials are required in particular to assess the risk of hyperkaliemia even if no clear evidence to support limiting plant-based foods in normokalemia patients. The evolution of the society and the realization of clinical studies on the topic in the coming years will be helpful to validate and motivate CKD patients to adopt this diet in safety conditions.”

  • “Line 69: You say "Numerous observational studies...", while you cite only one reference. Please add some more.”

Thank you for pointing this out, we have added “Haring et al. J Ren Nutr 2018” and “Oosterwijk et al. Kidney Int. Reports 2019”, reference 15 en 16.

  • “Line 72: I think that the sentence "In the latest recent meta-analysis.." would better start with a word showing the antithesis with the previous findings, like "however".”

We corrected this sentence as requested by the reviewer

Reviewer 3 Report

Dear authors,

i find the review very interesting and detailed. Different aspects of amminoacids composition and their effects on glomerular blood flow have been explored; some evidences are well known and others are novel.

I would like to underline an important point for a review; it's not described the method used to select literature and eventually the grading score. As the authors said, the topic is widely discussed and with some debated arguments. For this reasons, I think that clear selection process is useful to obtain stronger data.

Author Response

Dear authors,

  • I find the review very interesting and detailed. Different aspects of amminoacids composition and their effects on glomerular blood flow have been explored; some evidences are well known and others are novel

Thank you for this positive commentary.

  • “I would like to underline an important point for a review; it's not described the method used to select literature and eventually the grading score. As the authors said, the topic is widely discussed and with some debated arguments. For this reasons, I think that clear selection process is useful to obtain stronger data.”

For this review, we did not conduct a systematic review of the literature according to defined criteria, but we did select clinical trials that seemed to us to be the most robust and most relevant on the subject under review. We have selected in PubMed base for Table 1 only studies with a follow up over 6 months and for Table 2 we have only selected RCT in CKD patients treated with plant-based diet. We have added this point in the manuscript.

Round 2

Reviewer 3 Report

The work is useful to have a wide overview on plant-based diet in CKD. Some mechanisms have been fixed and others are still unknown. 

Data strenght is low, as i said in the previous review, but it's reported in the text.

Author Response

The work is useful to have a wide overview on plant-based diet in CKD. Some mechanisms have been fixed and others are still unknown.

Data strenght is low, as i said in the previous review, but it's reported in the text.

Thank you for this remarks.
We have underlight that indeed data strenght is low.

"It is important to mention that diet investigations are associated with several biases and it is important to consider that the association between socio-economic level and animal-protein intakes might influence the results of observational studies.
Therefore, the new international guidelines of the Kidney Disease Outcomes Quality Initiative (KDOQI) did not recommend plant protein over animal protein in CKD due to lack of evidence and because overall data strength is low. Yet, they do recommend a plant-based pattern in CKD patients for its potential benefits on body weight, blood pressure, and net acid production but not for GFR decline [6]."